# Association between Kinesiophobia and Knee Pain Intensity, Joint Position Sense, and Functional Performance in Individuals with Bilateral Knee Osteoarthritis

**DOI:** 10.3390/healthcare10010120

**Published:** 2022-01-07

**Authors:** Mastour Saeed Alshahrani, Ravi Shankar Reddy, Jaya Shanker Tedla, Faisal Asiri, Adel Alshahrani

**Affiliations:** 1Department of Medical Rehabilitation Sciences, King Khalid University, P.O. Box 960, Abha 61421, Saudi Arabia; msdalshahrani@kku.edu.sa (M.S.A.); jtedla@kku.edu.sa (J.S.T.); fasiri@kku.edu.sa (F.A.); 2Department of Physical Therapy, College of Applied Medical Sciences, Najran University, P.O. Box 1988, Najran 11001, Saudi Arabia; amsalshahrani@nu.edu.sa

**Keywords:** kinesiophobia, knee osteoarthritis, proprioception, joint position sense, functional performance

## Abstract

In current clinical practice, fear of movement has been considered a significant factor affecting patient disability and needs to be evaluated and addressed to accomplish successful rehabilitation strategies. Therefore, the study aims (1) to establish the association between kinesiophobia and knee pain intensity, joint position sense (JPS), and functional performance, and (2) to determine whether kinesiophobia predicts pain intensity, JPS, and functional performance among individuals with bilateral knee osteoarthritis (KOA). This cross-sectional study included 50 participants (mean age: 67.10 ± 4.36 years) with KOA. Outcome measures: The level of kinesiophobia was assessed using the Tampa Scale of Kinesiophobia, pain intensity using a visual analog scale (VAS), knee JPS using a digital inclinometer, and functional performance using five times sit-to-stand test. Knee JPS was assessed in target angles of 15°, 30°, and 60°. Pearson’s correlation coefficients and simple linear regressions were used to analyze the data. Significant moderate positive correlations were observed between kinesiophobia and pain intensity (r = 0.55, *p* < 0.001), JPS (r ranged between 0.38 to 0.5, *p* < 0.05), and functional performance (r = 0.49, *p* < 0.001). Simple linear regression analysis showed kinesiophobia significantly predicted pain intensity (B = 1.05, *p* < 0.001), knee JPS (B ranged between 0.96 (0° of knee flexion, right side) to 1.30 (15° of knee flexion, right side)), and functional performance (B = 0.57, *p* < 0.001). We can conclude that kinesiophobia is significantly correlated and predicted pain intensity, JPS, and functional performance in individuals with KOA. Kinesiophobia is a significant aspect of the recovery process and may be taken into account when planning and implementing rehabilitation programs for KOA individuals.

## 1. Introduction

Knee osteoarthritis (KOA) is a widespread chronic degenerative joint disease that burdens the public health system [1,2]. Around 250 million people worldwide suffer from this degenerative joint disease [3]. Females over 65, obese people, and African Americans have the highest risk of developing OA [4]. Patients with KOA seek treatment due to pain and functional limitations. More pain frequently means lower physical function, which means less ability to perform daily tasks, including walking short and long distances, climbing stairs, and sitting-to-stand [5]. Patients with knee OA have several conservative management options, including exercise therapy, weight loss, patient education, activity modification, footwear, bracing, and physical modalities [6,7,8,9]. 

Knee joint degeneration with pathological interactions and psychological factors can influence pain and physical dysfunction to a more considerable extent. It is well-established that subjects with chronic pain and disability have an increased fear of movement or harmful activity, and it is a primary cognitive factor that may cause anxiety and depression in KOA individuals. Therefore, physical therapy assessments should focus on pain-related fear, and other factors that may influence this fear, to better manage KOA patients with chronic pain, disability, and other musculoskeletal and psychological factors. 

Kinesiophobia is a fear of bodily movement and actions induced by a worry of painful injury or re-injury. Pain catastrophizing is a negative cognitive and affective response to expected or actual pain [8]. Individuals with chronic pain cannot disengage from unpleasant thinking and a sense of helplessness in dealing with pain. These habits may be adaptive in acute pain. Patients endure a vicious loop that maintains chronic pain and functional incapacity, worsening impairment, and pain perception thresholds.

Chronic KOA is a multifactorial disease, and patients present with increased pain, functional disability, and decreased quality of life [10]. Knee joint mobility is crucial in the elderly for maintaining functional independence. In subjects with KOA, specific movements in flexion and rotation cause increased pain levels [11]. This increased pain on movements causes fear of movement, and the individuals will tend to avoid these movements [12,13]. Avoiding activities and weight-bearing on the knee joint for an extended period will decrease knee muscle strength and endurance, especially in individuals with KOA [14,15]. Burgess et al. [16] demonstrated that impaired knee muscle performance would result in persistent knee degeneration and significantly impact functional performance and functional limitations [16]. Further, proprioception is an essential factor to maintain balance and stability in the elderly [17]. Fear of movement and catastrophic behavior in KOA individuals can affect the somatosensory system, impairing knee joint position sense (JPS).

In the elderly with KOA, progressive degeneration and deterioration of muscle strength, endurance, and proprioceptive impairments may be influenced by fear of movement [18]. Elderly individuals with a high level of kinesiophobia can significantly have increased pain levels and decreased JPS and functional performance with KOA. In addition, functional limitations, such as squatting, walking, performing sitting-to-standing actions, and climbing up and down the stairs, in the elderly are partially due to the normal aging process [19]. The presence of kinesiophobia may further deteriorate functional performance and impact the activities of daily living [20,21]. Even though kinesiophobia had been studied in different musculoskeletal conditions, its impact on knee pain intensity, knee JPS, and functional performance in elderly individuals with KOA is limited. Therefore, the objectives of this study are (1) to determine the association between kinesiophobia, pain intensity, knee JPS, and functional performance, and (2) to see if kinesiophobia predicts pain intensity, JPS, and functional performance in KOA individuals.

## 2. Methodology

### 2.1. Study Design and Participants

This cross-sectional study included 50 elderly adults with a diagnosis of KOA. The study was conducted between May and November 2021 in the department of medical rehabilitation sciences, King Khalid University, Abha, Saudi Arabia. All the subjects were referred to the university physical therapy clinic by an orthopedic or general physician with a diagnosis of unilateral or bilateral OA. Participants were included if they (1) were over the age of 40 years, (2) had knee pain for longer than three months, (3) were radiologically confirmed for the presence of OA changes in the tibiofemoral joint bilaterally, and 4) were able to understand and follow the commands of the examiner. The subjects were excluded if they had (1) a previous injury or surgery to the lower extremities, (2) a history of systemic inflammatory arthritis, (3) a history of meniscus or ligament injuries in the knee, and (4) infiltration corticosteroids in the knees in the last six months. 

This study followed the Declaration of Helsinki principles. The research ethics committee at King Khalid University (HAPO-06-B-001) reviewed and approved this study (ECM#2021-4504). Before participating in this study, all individuals provided written informed consent.

### 2.2. Outcome Measures

#### 2.2.1. Kinesiophobia

The Tampa Scale of Kinesiophobia (TSK) is a self-reported questionnaire that assesses fear of injury based on fear avoidance behavior and fear of activity [22]. TSK has 17 components. Each scale runs from one (strongly disagree) to four (strongly agree). The responses are added together to get a total score, with higher values indicating greater pain-related fear. The total score ranges between 17 and 68, with 17 indicating no kinesiophobia, 68 indicating severe kinesiophobia, and 37 indicating the presence of kinesiophobia.

#### 2.2.2. Knee Pain Intensity-Visual Analogue Scale (VAS)

The current level of knee pain intensity was evaluated on a 0 to 100 mm continuous VAS anchored by two statements: "0" meaning no pain, and "100" meaning the worst imaginable pain. Thus, the individuals make a mark on the scale, which indicates their current pain intensity level. Previous authors used VAS to measure pain intensity in different musculoskeletal conditions and it has demonstrated good to excellent reliability [23].

#### 2.2.3. Knee Joint Position Sense

A dual digital inclinometer (Dualer IQ JTech Medical, Salt Lake City, UT, USA) was used to test knee joint position sense. The degree of joint position error (JPE) was recorded in degrees. The dual inclinometer consists of the primary and auxiliary inclinometer. The accessories include three velcro straps, an assembly wire connecting the inclinometers, and two metal plates to align the inclinometer unit. The digital inclinometer had a significant level of validity compared with an isokinetic dynamometer (ICC = 1.0, SEM = 1.39, *p* < 0.001) and also showed excellent intra- and inter-tester reliability for measuring JPS (ICC = 0.994, SEM = 1.67, *p* < 0.001) [24,25,26].

Individuals sat on the edge of a bed with their eyes closed. For the evaluation of knee JPS, the passive to active joint repositioning approach was adopted. The inclinometer unit was calibrated to zero at the start of the experiment by placing it on a flat surface. The primary inclinometer was placed and secured using velcro at the lower lateral surface of the femur along the joint line (Figure 1). The auxiliary inclinometer was placed lower, to the fibula’s head, and secured with velcro. An ethernet cable connects the inclinometers to complete the circuit. The subjects were asked to reposition to the target angles at 15°, 30°, and 60°. Next, the examiner passively flexed the subject’s knee to the target angle (15°, 30°, and 60°) and held it for 5 s. The subject was asked to remember this position, and then the knee was extended to the neutral position. The subject was then asked to reposition to the target angle and any deviation from this target angle was recorded as reposition accuracy measured in degrees. Each knee JPS test was repeated three times and the average of the three trials was used for analysis.

#### 2.2.4. Functional Performance

The five times sit-to-stand (5STS) [27] test measured functional performance in subjects with knee OA. The individual sits against the chair with arms folded across the chest.

It is a timed test, and the individuals are instructed to rise to a standing position from the seated position (43 cm high armless chair) five times as quickly as possible [27]. This test showed excellent reliability (ICC = 0.95), and it is a valid measure of functional mobility in older individuals [28]. The knee JPS test was performed three times and the best (lowest) value was used to compute the analysis.

### 2.3. Statistical Analysis

The statistical analyses were performed using the "Statistical Package for Social Sciences" (SPSS) Version 24.0 (SPSS Inc., Chicago, IL, USA). The Shapiro–Wilk test examined the normal distribution of the study variables. For each of the study measures, descriptive statistics were computed. Means and standard deviations (SD) were used to represent quantitative variables. The association between kinesiophobia, pain intensity, knee JPS, and functional performance was analyzed using Pearson correlation coefficients. Simple linear regression analysis was performed to assess how kinesiophobia predicts pain intensity, JPS, and functional performance. The regression model included knee pain intensity, JPEs and functional performance, and TSK scores. In all instances, a *p* ≤ 0.05 significant threshold was considered.

## 3. Results

A total of 106 people were initially recruited, with 33 being excluded throughout the screening process. Sixteen of them were under the age of 40 years, and 17 had previously suffered ligament and meniscus injuries. As a result, 73 people were invited to our study lab for a clinical evaluation and their knees were radiographed. One participant had osteochondroma, 13 did not exhibit OA changes during radiological evaluation, and nine had positive ligament tests; thus, they were eliminated. The final study sample consisted of 50 people, and their demographic characteristics, pain intensities, target JPS, and functional performance values are shown in Table 1. The Shapiro–Wilk test showed the normal distribution of study data (*p* > 0.05). 

Correlations between kinesiophobia and pain intensity, knee JPS at different target angles, and functional performance are summarized in Table 2 and Figure 2. 

The results of Pearson correlation coefficients (r) showed a significant moderate positive correlation between kinesiophobia and knee pain intensity (r = 0.55, *p* < 0.001). In addition, kinesiophobia showed a significant positive moderate association with knee JPS at the target angles of 15°, 30°, and 60° (*p* < 0.05). The results indicate that an increase in kinesiophobia increases knee joint position errors. The most significant association was between kinesiophobia and right knee JPS at 15° of flexion (r = 0.50, *p* < 0.001). Also, there was a moderate positive association between kinesiophobia and functional performance (r = 0.49, *p* < 0.001).

The kinesiophobia significantly predicted knee pain intensity (B = 1.05, *p* < 0.001), knee JPS (B ranged between 0.96 (30° of knee flexion for right side) and 1.30 (15° of knee flexion for right side)), and functional performance (B = 0.57, *p* < 0.001). Table 3 summarizes the results of the linear regression study.

## 4. Discussion

This study aimed to determine the association between kinesiophobia, pain intensity, knee JPS, and functional performance, and see if kinesiophobia predicts pain intensity, JPS, and functional performance in KOA individuals. The results of the study show that kinesiophobia is positively associated with knee pain intensity, knee JPS, and functional performance, indicating that, as kinesiophobia increases, knee pain intensity increases and the JPS and the functional performance decrease. Further, kinesiophobia significantly predicted pain intensity, knee JPS, and functional performance in individuals with KOA. 

In this study, individuals with KOA displayed a fear of movement as measured by the Tampa Scale of Kinesiophobia. The fear of movement seems to be a coping strategy to explain the increased level of knee pain. Further, in the regression analysis, kinesiophobia significantly predicted pain intensity level. These findings in our study are in accordance with studies that showed a significant association between kinesiophobia and pain intensity level in different conditions with musculoskeletal pain [20,29,30]. A recent study reveals that fear of movement and catastrophic thoughts lead an individual to painful adverse consequences and affect the neurophysiology of pain regulation [31]. The functional MRIs of chronic pain individuals with increased fear of movement revealed increased activity in cortical areas related to attention, anticipation, and emotional components of pain [31]. Different authors have seen an association between fear avoidance behavior and pain in the lower back and those with rheumatic conditions [32,33,34,35]. However, other researchers have found no such relation [32,36,37]. These results cannot be compared to our study as the populations recruited and the study methods adopted differ. 

Our study supports the findings that higher kinesiophobia can cause poor knee JPS in subjects with KOA, confirming our hypothesis that joint position and its behavior are linked to fear of movement. It infers that muscles around the knee significantly contribute to knee joint proprioception and force-generating capabilities, and kinesiophobia can influence these factors and modify the afferent input contributing to altered knee JPS [38]. Pakzad et al.’s [39] study showed that fear of movement altered muscle activation patterns and motor control [39]. Individuals with chronic pain with OA changes also may exhibit a similar association. Limited studies show the association between kinesiophobia and knee proprioception, but studies in other regions with musculoskeletal conditions have shown a positive association. Asiri et al. [20] showed mild to moderate positive correlations between kinesiophobia and cervical joint position errors (extension: r = 0.48, *p* < 0.001; right rotation: r = 0.31, *p* = 0.011) [20]. Similar to our study results, the regression analysis proved that kinesiophobia was a significant predictor of JPS [20]. Aydoğdu et al. [40] did not find any relation between kinesiophobia and proprioception in individuals following anterior cruciate ligament reconstruction. It may be that the subjects in Aydoğdu et al.’s study had lower kinesiophobia scores than those in this study. Our study’s subjects had a mean kinesiophobia score of 48.68 ± 4.38, compared to Aydoğdu et al.’s study’s subjects’ mean score of 36.54 ± 4.22 [40]. In addition, we have measured JPS at 15°, 30°, and 60° of knee flexion, and we have chosen these angles to represent more functional positions. Maybe the findings would be different if we had chosen other angles with increased knee flexion (e.g., 100° or 135°).

This study also showed a positive association with functional performance. Similar investigations have discovered a strong association between kinesiophobia and decreased range of motion and increased pain intensity. According to some authors, long-term kinesiophobia can lead to muscle fatigue and disuse atrophy, create a vicious loop, and impair functional performance [41,42]. Induced by pain stimulation and muscle weakness, kinesiophobia eventually results in fear-avoidance behaviors, which can have a negative impact on functional performance in the lower limbs [21]. Kinesiophobia significantly predicted functional performance in KOA individuals. The current study’s findings are congruent with those of a prior study [43], which indicated that TSK score was positively correlated with walking impairment, and kinesiophobia significantly predicted functional performance in subjects with chronic pain [20].

From a rehabilitation perspective, it is critical to distinguish between functional limitations caused by pain and fear-avoidance behaviors. Screening approaches may be beneficial in identifying plausible risk factors and identifying people at risk, ideally during the first phase of pain. The individuals in the present study who had a high level of kinesiophobia had increased pain intensity and decreased JPS and functional performance. Therapists should benefit from employing questionnaires as a screening tool before initiating planned treatment. Screening for kinesiophobia is critical for designing an effective rehabilitation program, and utilizing simple questionnaires, such as the TSK, enables the identification of pain patients with elevated kinesiophobia scores. Additionally, therapists must have the ability to identify patients who require additional treatment for psychological distress. Further, the outcomes of this study emphasize the importance of using a more holistic approach when assessing and managing patients with chronic pain.

### Limitations of the Study

This cross-sectional study included a sample of 50 individuals, which is relatively small, and these findings can’t be generalized to all KOA individuals. We evaluated five times sit-to-stand as a measure of functional performance. Additionally, other outcomes, such as walking endurance, speed tests, and back endurance, may be recommended to obtain a more comprehensive functional performance analysis. Furthermore, as age increases, muscle mass, strength, and proprioceptive functionality decrease [44,45,46]. As a result, the loss in JPS and functional performance shown in this study could be due to aging processes, rather than kinesiophobia. We did not control for age, formal education, leisure time activities, smoking, and sleeping hours, which could have confounded the study results. Future research should examine these confounding variables and their impact on the outcomes.

## 5. Conclusions

In conclusion, kinesiophobia correlated significantly with pain intensity, JPS, and functional performance. Further, kinesiophobia significantly predicted pain intensity, JPS, and functional performance in individuals with KOA. Kinesiophobia is a significant aspect of the recovery process and may be considered when planning and implementing rehabilitation programs for KOA individuals. 

## Figures and Tables

**Figure 1 healthcare-10-00120-f001:**
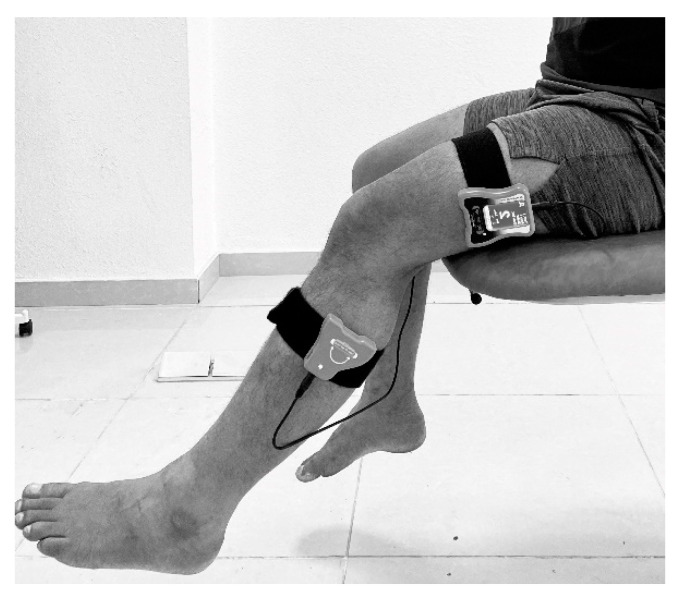
Testing knee joint position sense.

**Figure 2 healthcare-10-00120-f002:**
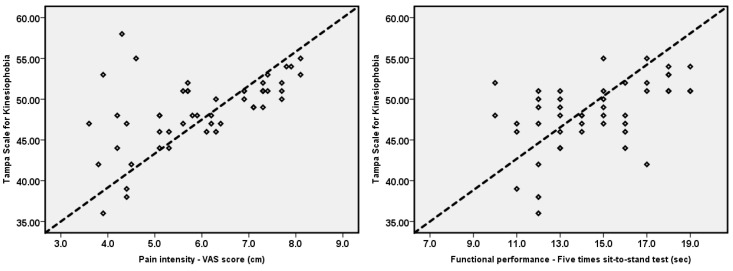
Association between kinesiophobia and pain intensity, functional performance, and JPS at target angles of 15°, 30°, and 60° for left and right sides.

**Table 1 healthcare-10-00120-t001:** Characteristics of the study participants (*n* = 50).

Variables	Mean ± SD
Age (years)	67.10 ± 4.36
BMI (kg/m^2^)	29.59 ± 4.52
VAS (cm)	06.00 ± 1.34
TSK total score	48.68 ± 4.38
Knee JPS angles	
15° of knee flexion—left	4.08 ± 1.74
15° of knee flexion—right	5.14 ± 1.70
30° of knee flexion—left	6.04 ± 1.73
30° of knee flexion—right	4.22 ± 1.59
60° of knee flexion—left	5.54 ± 1.67
60° of knee flexion—right	6.56 ± 1.68
Functional performance—five times sit-to-stand test (s)	15.30 ± 3.76

BMI = body mass index, VAS = visual analog scale, TSK = Tampa Scale of Kinesiophobia, JPS = joint position sense.

**Table 2 healthcare-10-00120-t002:** Association between kinesiophobia, pain intensity, functional performance, and proprioception (*n* = 50).

Correlation Variables	Kinesiophobia–TSK Score	
	r	*p* Value
Knee pain intensity-VAS (mm)	0.55	<0.001
Knee JPS angles		
15° of knee flexion—left	0.48	<0.001
15° of knee flexion—right	0.50	<0.001
30 ° of knee flexion—left	0.38	<0.001
30 ° of knee flexion—right	0.42	0.003
60 ° of knee flexion—left	0.46	0.001
60 ° of knee flexion—right	0.42	0.002
Functional performance—five times sit-to-stand test (s)	0.49	<0.001

VAS = visual analog scale, JPS = joint position sense. The correlation was tested using Pearson’s correlation coefficient analysis.

**Table 3 healthcare-10-00120-t003:** Simple linear regression of TSK scores and explanatory variables (*n* = 50).

Variable	B	SE	t-Value	*p* Value
Knee pain intensity-VAS (mm)	1.80	0.39	4.58	<0.001
Knee JPS angles				
15° of knee flexion—left	1.21	0.32	3.77	<0.001
15° of knee flexion—right	1.30	0.32	4.04	<0.001
30° of knee flexion—left	0.96	0.34	2.84	0.007
30° of knee flexion—right	1.14	0.36	3.17	0.003
60° of knee flexion—left	1.20	0.34	3.58	0.001
60° of knee flexion—right	1.10	0.34	3.22	0.002
Functional performance—five times sit-to-stand test (s)	0.57	0.15	3.87	<0.001

JPS = joint position sense, B = unstandardized coefficients, SE = standard error.

## Data Availability

On request to the corresponding author, Ravi Shankar Reddy (rshankar@kku.edu.sa), all data are available at the Department of Medical Rehabilitation Sciences.

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
