# Peer review of "Association between Kinesiophobia and Knee Pain Intensity, Joint Position Sense, and Functional Performance in Individuals with Bilateral Knee Osteoarthritis"

_healthcare, 2022, doi:10.3390/healthcare10010120_

Round 1

Reviewer 1 Report

Manuscript ID: healthcare-1516827

Manuscript title: Association between kinesiophobia and knee pain intensity, joint position sense, functional performance in individuals 3 with bilateral knee osteoarthritis

Comments

This study investigated the relationship between kinesiophobia, pain intensity, knee joint position sense, and functional performance in people with knee osteoarthritis. The manuscript is well written and organized, although it lacks fundamental information that precludes the full assessment of the report. Please refer to the comments below.

Major comments

  1. Main text, Statistical analysis. There is no description of the regression analysis. The authors should report, among other items: what variables entered the model as predictors or outcomes; how predictors entered the model (force-entered, backward selection?); whether missing data was present and how it has been handled if present; what measures of goodness-of-fit were calculated.

  1. Results. Because there is no information regarding the regression models, it is not possible to understand Table 3. It is apparent that r² values are indeed R² (goodness-of-fit of the model) because these values in Table 3 are not consistent with squared values of the Pearson’s correlation coefficient for each predictor-outcome relation in Table 2. Therefore, it is apparent that several independent models were tested, with has not much additional information than the Pearson’s correlation.

  1. Discussion cannot be assessed due to the lack of detailed information regarding the statistical analysis.

Minor comments

  1. Abstract. From the description of the results, it is unclear whether a single model with all three variables (pain intensity, joint position sense, functional performance) or three independent models was tested. Also, please use β (beta) for regression coefficients and R² (capital r-squared) when referring to the goodness-of-fit of the model, respectively.

  1. Abstract and main text. Please revise the total number of participants (50 in the abstract, 56 in the main text/methods section, 50 again in the main text/results section).

Author Response

1.       

Main text, Statistical analysis. There is no description of the regression analysis. The authors should report, among other items: what variables entered the model as predictors or outcomes; how predictors entered the model (force-entered, backward selection?); whether missing data was present and how it has been handled if present; what measures of goodness-of-fit were calculated.

Results. Because there is no information regarding the regression models, it is not possible to understand Table 3. It is apparent that r² values are indeed R² (goodness-of-fit of the model) because these values in Table 3 are not consistent with squared values of the Pearson’s correlation coefficient for each predictor-outcome relation in Table 2. Therefore, it is apparent that several independent models were tested, with has not much additional information than the Pearson’s correlation.

·         Apologies for the confusion.

·         We have computed the simple regression analysis again and made changes to the table 3.

·         The regression model included knee pain intensity, JPE’s and functional performance and TSK scores

·         We did not select any method (neither forward or backward).

·         Model-fit was selected, estimates and confidence intervals were selected as a measure of regression coefficients.

·         Simple linear regression analysis was performed to assess how kinesiophobia predicts pain intensity, JPS, and functional performance.

·         Regression allows us to estimate how a dependent variable changes as the independent variable(s) change.

·         We used B= Unstandardized Coefficients, SE = standard error, t-value and p value to represent the results.

·         Unstandardized coefficients are used to interpret the effect of each independent variable on the outcome.  

·         The standard error of the regression (SE) represents the average distance that the observed values fall from the regression line.

·         The t-value measured the size of the difference relative to the variation in our sample data.

·         I am not sure about the goodness-of-fit of the model

·         There was no missing data

2.       

Discussion cannot be assessed due to the lack of detailed information regarding the statistical analysis.

·         Accepted.

3.       

Abstract. From the description of the results, it is unclear whether a single model with all three variables (pain intensity, joint position sense, functional performance) or three independent models was tested. Also, please use β (beta) for regression coefficients and R² (capital r-squared) when referring to the goodness-of-fit of the model, respectively.

·         Simple linear regression analysis was performed to assess how kinesiophobia predicts pain intensity, JPS, and functional performance.

·         All the variables were tested independently

4.       

Abstract and main text. Please revise the total number of participants (50 in the abstract, 56 in the main text/methods section, 50 again in the main text/results section).

·         Sorry for the typo error. This study included 50 participants only.

Reviewer 2 Report

BRIEF SUMMARY

This was a cross-sectional study with the aim to establish the association between kinesiophobia and knee pain intensity, joint position sense (JPS), and functional performances in people with knee OA. Authors conclude that Kinesiophobia is significantly correlated and predicted pain intensity, JPS, and functional performance in individuals with knee OA.

I congratulate authors on their work. This is a well-written and structured paper with informative figures and tables. The topic is timely and clinically important. However, before it can be published, I suggest authors to consider my points below.

SPECIFIC COMMENTS

ABSTRACT

Line 25: ‘’kinesiophobia SHOULD be monitored”: please tone down your conclusions. Although you found some associations, the study design and lack of consideration of potential confounding factors prevent, in my opinion, from coming to such strong conclusions.

INTRODUCTION

Line 38: I think the reader would benefit from knowing what are the established interventions in people with knee OA, especially in terms of the assessed outcomes. Some examples to cite: lifestyle-modification (https://pubmed.ncbi.nlm.nih.gov/11567539/ ), exercise (https://www.ncbi.nlm.nih.gov/pmc/articles/PMC3635671/ ), physical modalities ( https://pubmed.ncbi.nlm.nih.gov/25162407/ ) knee orthoses (https://pubmed.ncbi.nlm.nih.gov/29931372/) etc.

In the current form, it is quite difficult to figure out from the information flow in the introduction, why it is important to study this, what is the added value of this paper to current knowledge, and who will benefit from this. Please clarify.

Lines 73-75: Please check and rephrase. Did you mean: the knowledge regarding its impact….. is limited?

METHODS

You tend to interchangeably use the words correlation, relationship, association. Please use one for consistency or explain the difference between them (whole manuscript)

Please provide details regarding the validity and reliability of the tool for measuring JPS.

Line 127: “thrice” ?

Statistics:

  1. How did you address statistical bias due to the issue of multiple testing?
  2. ‘”Raw data and percentages are used to express qualitative variables.”: did you actually have any?
  3. You claim you assessed the normality of the data. If so, please provide results in a relevant section
  4. You mention how you tested associations (Pearson) but you did not state what statistical tests you used to establish prediction.

RESULTS

Very well presented

DISCUSSION

I suggest starting off the discussion by reminding the reader about the study objectives.

The discussion lacks a paragraph regarding the clinical implications of the findings. For example, you could have discussed potential treatments for kinesiophobia.

Furthermore, the variables such as age, formal education, leisure time activities, smoking, and sleeping hours are not considered as these can confound the study results.” That is actually the opposite – you should have considered them and adjust the regression models for such potential confounders to confirm the association exists or is just due to other factors. As such, if you did not collect data regarding these variables, this sentence should be rephrased to “We did not control for age, formal education, leisure time activities, smoking, and sleeping hours which could have confound the study results”.

CONCLUSION

Please ensure conclusions are similar to the one presented in the abstracts.

Author Response

Thank you for your effort and time in reviewing our manuscript. The reviewing process has significantly improved the quality of this manuscript. Therefore, I am submitting this "Response to reviewers" document summarizing the changes we made in response to the critiques.

Round 2

Reviewer 1 Report

Thank you for submitting a revised version of your manuscript covering my previous comments. All comments have been properly addressed. I have no new comments.

Author Response

Thank you

Reviewer 2 Report

The authors did not address some of my comments appropriately.

  1. The conclusions are still overestimated. You should avoid using "should" and instead use "may/might" given your study design (not an RCT) and lack of consideration of potential confounding.
  2. Line 40. You provide the wrong references which are highly irrelevant. The reference I have suggested should be provided.

Also, instead of simply saying in your cover letter that things have been done, you should have tried to facilitate the work of a reviewer by providing line numbers and a manuscript version with track changes.

Author Response

Response to Reviewer comments

Thank you for your effort and time in reviewing our manuscript. The reviewing process has significantly improved the quality of this manuscript. Therefore, I am submitting this "Response to reviewers" document summarizing the changes we made in response to the critiques.

Reviewer 2

Sl.no

Queries

Response to queries

Changes made in the manuscript.

1

The conclusions are still overestimated. You should avoid using "should" and instead use "may/might" given your study design (not an RCT) and lack of consideration of potential confounding.

The conclusion is modified.

The word “should” is replaced with “may.”

·         Line 26 (Abstract)

·         Line 283 (main text conclusion)

2

Line 40. You provide the wrong references which are highly irrelevant. The reference I have suggested should be provided.

The references you have suggested are considered.

·         O'Reilly, S.; Doherty, M. Lifestyle changes in the management of osteoarthritis. Best Practice & Research Clinical Rheumatology 2001, 15, 559-568.

·         Vincent, K.R.; Vincent, H.K. Resistance exercise for knee osteoarthritis. Pm&r 2012, 4, S45-S52.

·         Cherian, J.J.; Kapadia, B.H.; Bhave, A.; McElroy, M.J.; Cherian, C.; Harwin, S.F.; Mont, M.A. Use of transcutaneous electrical nerve stimulation device in early osteoarthritis of the knee. The journal of knee surgery 2015, 28, 321-328.

·         Cudejko, T.; Van Der Esch, M.; Schrijvers, J.; Richards, R.; Van Den Noort, J.C.; Wrigley, T.; Van Der Leeden, M.; Roorda, L.D.; Lems, W.; Harlaar, J. The immediate effect of a soft knee brace on dynamic knee instability in persons with knee osteoarthritis. Rheumatology 2018, 57, 1735-1742.

·         Line 40
